# Adolescents’ Explanatory Models for Headaches and Associations with Behavioral and Emotional Outcomes

**DOI:** 10.3390/children8030234

**Published:** 2021-03-18

**Authors:** Verena Neß, Anna Könning, Gerrit Hirschfeld, Julia Wager

**Affiliations:** 1German Paediatric Pain Centre, Children’s and Adolescents’ Hospital, 45711 Datteln, Germany; a.kupitz@deutsches-kinderschmerzzentrum.de (A.K.); j.wager@deutsches-kinderschmerzzentrum.de (J.W.); 2Department of Children’s Pain Therapy and Paediatric Palliative Care, Faculty of Health, School of Medicine, Witten/Herdecke University, 58448 Witten, Germany; 3CareTech OWL Center for Health, Welfare and Technology, Faculty of Business, University of Applied Sciences Bielefeld, 33619 Bielefeld, Germany; gerrit.hirschfeld@fh-bielefeld.de

**Keywords:** headaches, biopsychosocial factors, pediatrics, causal explanations, concordance

## Abstract

More than one-third of adolescents experience recurrent headaches. Usually, these headaches are of primary origin and modulated by different biological and psychosocial factors. While parents are often consulted in scientific research and medical care about the nature of their child’s headache, it is unclear to what extent parents and their children agree upon the factors that cause them. Adolescents’ own attributions of headaches have rarely been investigated, and it is unclear how those attributions affect behavioral and emotional outcomes. In the present study, adolescents with chronic headaches (*N* = 248) and their parents (*N* = 120) rated the influence of various biological and psychosocial factors on the adolescents’ headaches. Associations between these factors and several behavioral and emotional outcomes were examined. The most frequently reported factor by both samples was stress; however, concordance between parents and adolescents was generally low. The factor “other disease” was significantly associated with medication consumption and school absence. This study is one of the first to provide insights into adolescents’ own attributions of headaches. Furthermore, the significant associations of the factor with behavioral outcomes reveal the importance of understanding personal explanatory models of headache. Future studies should examine associations between subjective headache causes and the individual’s experience of the disorder to improve headache interventions.

## 1. Introduction

Headaches are a common health problem in children and adolescents. By the age of 15, approximately 70–75% of adolescents will have experienced significant headaches [1,2]. Around one-third of adolescents report recurrent headaches [3]. There are manifold types of headaches that are categorized into two major groups [4,5], namely primary headache and secondary headache. The most common primary headaches are tension-type headache and migraine. Secondary headaches occur due to underlying diseases; commonly infection, but rarely tumors or other malignant conditions [6]. Chronic and recurrent headaches in children and adolescents are usually a primary condition [7].

Chronic headache is a burdensome experience that is often related to further negative outcomes. In general, suffering from headaches often leads to absence from school, impaired school functioning, or restrictions in activities [5,8,9]. Around 20% of adolescents suffering from chronic headache report medication overuse [10]. Moreover, headaches could be associated with psychiatric comorbidities like depression and anxiety disorders [11,12]. Headaches are also often found in adolescents with somatic symptom disorder [13], a condition in which a physical symptom like pain is associated with serious functional or emotional impairment [14].

Regarding the understanding of illness and chronic pain, there has been a shift in the experts’ perspective over the last decades from a purely biomedical view to biopsychosocial approaches [15,16]. Various biopsychosocial factors could already be associated with the occurrence of primary headaches [6]. For example, it has been found that there is a strong genetic predisposition for migraine [17,18]. Tension-type headaches are associated with constant increased muscular tension [6,19,20]. Stress, too, is an important psychosocial factor in triggering or worsening headaches [21,22]. Furthermore, psychosocial factors like high school demands or conflicts with peers were found to increase the occurrence of headaches in adolescents [23,24].

To better understand people’s concepts of pain, several studies have already examined mothers’ and teachers’ attributions of children’s chronic pain [25,26]. While the majority of mothers understood the biopsychosocial causes of their child’s chronic pain, around two-thirds of teachers had either a purely physical or psychological understanding of chronic pain. Nevertheless, both mothers and teachers rated stress as one of the dominant psychological causes of chronic pain. The most frequently mentioned physical causes were nerve damage (teachers) and sensitive organs (mothers). Furthermore, one recent study developed and validated the Concept of Pain Inventory [27], a questionnaire measuring general beliefs, knowledge, and perceived function of pain, as well as biological pain processes. However, it does not specifically focus on a child’s attribution of factors relevant to his or her own chronic pain condition. 

Taken together, adolescents’ own opinions and causal attributions for chronic or recurrent pain and headaches are poorly investigated. In addition, the level of agreement between parents and their children regarding the subjective attribution of factors causing chronic headaches remains unclear. Furthermore, it is not known if one’s own concept of explanatory factors of headaches is associated with outcomes like those mentioned.

This study aims to gain a deeper understanding of the individual view of factors associated with adolescents’ chronic headaches. First, we describe which biological and psychosocial factors adolescents and their parents consider relevant for chronic headaches and investigate their concordance. Second, we identify associations of these factors with various outcomes (medication use, days absent from school, disability, depression, and anxiety).

## 2. Materials and Methods

### 2.1. Study Design and Procedure

The present study was part of the “Chronic headache in adolescence: The patient perspective on healthcare utilization” (CHAP) project, a longitudinal study with five assessments each three months apart (T_1_–T_5_). For this analysis, only data from the fourth assessment (T_4_) were used, which focused on the adolescents’ and parents’ explanatory model of headaches. Data collection for this assessment was carried out as an online survey between July and December 2018. Study participants were recruited at five secondary schools in North-Rhine Westphalia, including all three German school types (Gesamtschule, Realschule, Gymnasium). Students between 10 and 18 years of age enrolled in the fifth to tenth grades—and their parents—were eligible. Only one parent participated in the study. All participants received detailed written and verbal information. To be included in the study, students and their parents needed to provide informed consent. After completing the survey, participants were compensated with a EUR 2 voucher for an online store. 

Ethics approval was deemed by the committee of Witten/Herdecke University (reference number 40/2017).

### 2.2. Sample 

Out of 3324 adolescents available at the five participating schools, *N* = 2280 students (52.2% female; *M*_age_ = 13.0, *SD* = 1.8; see [25] for detailed descriptions of exclusion criteria) and *N* = 1464 parents (81.1% mothers, 17.9% fathers, 1% others) were included in the study at T_1_. At T_4_, *N* = 1674 students and *N* = 1129 parents participated. Of the entire sample at T_4_, 15% (*N* = 248) of adolescents had chronic headache; only these were included in analyses. In addition, the corresponding data for 120 parents (82.5% mothers, 17.5% fathers) were used.

### 2.3. Measurements and Variables

Students were classified with chronic headache if the headaches were already present at the prior measurement point T_3_ (or at T_2_, if T_3_ data were missing), had occurred at least weekly in the prior three months, and had occurred during the last four weeks [28]. 

Factors potentially causing headaches are displayed in Table 1. Adolescents and parents were asked to rate each factor as “yes” (factor is relevant for the occurrence of headache) or “no” (factor is not relevant for the occurrence of headache). For this, adolescents and parents were asked the following: “What do you think your pain/your child’s pain is related to?” Factors were extracted from a prior qualitative study of adolescents with recurrent headaches and their parents who visited a primary care pediatrician. In the study, *N* = 18 adolescents and *N* = 17 parents participated in a semistructured interview where they reported which factors were relevant to the development and maintenance of the adolescent’s headaches. Those answers were analyzed using qualitative content analysis, and 12 categories were extracted. Those categories were used for the survey in this study. For the analysis of individual pain concepts, each option was categorized as biological or psychosocial. Biological factors included organic causes, nutrition, injury, another disease, puberty, predisposition, and poor exercise. Psychosocial factors comprised stress, high media consumption, conflicts in the family environment, high school demands, and conflicts with friends and/or peers. Each factor was assessed in a checklist, following the question specified above, and was not further specified in the survey. For example, the factor “nutrition” includes every relevant aspect that adolescents and their parents think might cause headaches. Similarly, the factor “other disease” includes any other diseases that the students and parents consider a causal explanation for headache. Furthermore, the factor “stress” comprises various stressors that students and parents may attribute to headaches, without further specification (i.e., undifferentiated stress).

In addition, we examined whether adolescents and parents had an exclusively biological, exclusively psychosocial, a combined biopsychosocial, or no pain concept (Table 1). Those who exclusively considered biological factors relevant to their headache were categorized as having a biological pain concept. Similarly, adolescents and parents who selected only psychosocial factors were categorized as having a psychosocial pain concept. Those who selected both biological and psychosocial factors were categorized as having a biopsychosocial pain concept. Finally, adolescents and parents who did not select any factor and who reported no causal explanation for the origin of their children’s pain were categorized as having no pain concept. 

Furthermore, students reported various outcome variables (see Table 2). These outcomes were assessed to examine potential associations with the factors they considered causal for their headaches. In the survey, students specified their medication consumption during the last four weeks. This information was transformed into a binary outcome variable by splitting consumption into “<4 days” and “≥4 days” within the last four weeks. In line with a prior study [29], this cut-off was chosen to identify those adolescents with high medication use (at least once a week on average) and differentiate them from students with unique medication intake. 

Moreover, adolescents reported the absolute number of days absent from school during the last four weeks. This variable was also split into two categories (<2 days absent, ≥2 days absent). The cut-off at two days was based on the established chronic pain grading categories by Wager et al. [30].

The validated German version of the Pediatric Pain Disability Index (PPDI; [31]) served to measure pain-related impairment. The PPDI consists of 12 items that are rated on a 5-point Likert scale (1 = never, 5 = always). The sum of all items reflects the severity of pain-related impairments in daily activities, where a higher score indicates greater impairment. For the present study, the PPDI variable was coded binarily. The binary categorization into “low disability” and “high disability” was guided by the chronic pain grading categories by Wager et al. [30]; a cut-off was set at 35. 

Furthermore, Major Depressive Disorder and Generalized Anxiety Disorder subscales of the Revised Children’s Anxiety and Depression Scale (RCADS; [32,33]) were assessed for each participant. In this questionnaire, participants rated corresponding items on a 4-point Likert scale (0 = never, 3 = always). The depression subscale consists of 10 items, and a maximum of 30 points can be achieved. The anxiety subscale has six items adding up to a possible maximum score of 18 points. The binary categorization of RCADS scores is previously described in Wager et al. [28,34], adapted from Chorpita et al. [34], to split groups into “no depressive/no anxiety symptoms” (no depression: <11, no anxiety: <7) and “depressive/anxiety symptoms” (depression: ≥11, anxiety: ≥7). 

### 2.4. Statistical Analysis

Descriptive statistics of frequencies and percentages were used to describe causes of headaches as rated by students and parents. Cohen’s Kappa was used to measure consensus between the adolescents’ and parents’ ratings according to the following criteria: 0, poor; 0 to 0.20, slight; 0.21 to 0.40, fair; 0.41 to 0.60, moderate; 0.61 to 0.80, substantial; and 0.81 to 1.00, (almost) perfect [35]. 

Next, least absolute shrinkage and selection operator (lasso) regressions [36] were conducted to investigate the relationship between each factor rated by adolescents and the following outcome variables: medication use (no: <4, yes: ≥4), school absence (no: <2 days, yes: ≥2 days), PPDI (below cut-off: ≤35, above cut-off: >35), RCADS scores for depression (no depression: <11, depression: ≥11), and RCADS scores for anxiety (no anxiety: <7, anxiety: ≥7). Lasso regression was chosen due to the large number of predictor variables able to be selected as causes of headaches. By using lasso regression, only meaningful predictors remain in the model, while less important predictors are set to zero. After running a ten-fold cross-validation on the data (seed = 874), the minimum mean cross-validated error was calculated individually for each outcome variable and defined as the penalty parameter λ (lambda). The penalty parameters were as follows: medication use = 0.022; school absence = 0.017; PPDI = 0.010; RCADS_Depression = 0.009, and RCADS_Anxiety = 0.025. 

Descriptive statistics and kappa tests were performed with SPSS Statistics software (Version 27 for Windows). The lasso regression was run with R software (Version 4.0.3; [37]). For the lasso analysis, the *glmnet* package [38] was used. Statistical significance was tested using the *selectiveInference* package [39].

## 3. Results

### 3.1. Demographics and Descriptives

Of the adolescent sample (*N* = 248), 195 participants were girls (78.6%), and 53 adolescents were boys (21.4%). The mean age was 13.4 (*SD* = 1.85) years. 

The distribution of the outcome variables is presented in Table 2. 

### 3.2. Factors Considered Causal for Headaches 

Adolescents’ ratings regarding factors causing their headaches are displayed in Table 1. The ratings revealed that more than half of the students rated stress as causing their headaches (*n* = 132; 53.4%). High school demands were rated second highest by *n* = 94 (38.1%) adolescents. Around one-fourth of all students (*n* = 63; 25.5%) considered puberty as causal for their headaches. The least frequently reported factors were poor exercise (*n* = 27; 10.9%) and organic causes (*n* = 28; 11.3%).

Factor ratings were available for *N* = 120 parents. Like the adolescents’ ratings, parents rated stress as the most common cause of the adolescents’ headaches (*n* = 51; 42.5%), followed by puberty (*n* = 49; 40.8%). Headaches caused by high school demands were noted by *n* = 38 (31.7%) parents. The most infrequent factors were organic causes (*n* = 4; 3.3%) and conflict with family (*n* = 9; 7.5%).

Cohen’s Kappa indicated that agreement regarding the ratings for the origin of headaches was generally low, ranging from poor agreement to fair agreement. However, there was reasonable agreement between parents and adolescents for puberty (κ = 0.382), poor exercise (κ = 0.322), and family conflict (κ = 0.282). All kappa values are displayed in Table 1. 

### 3.3. Distribution of Superordinate Pain Concepts

Superordinate pain concepts were defined for each adolescent and parent, based on factors considered causal for adolescents’ headaches. For this, each factor was categorized as biological or psychosocial (Table 1). This classification of pain concepts revealed that *n* = 128 (51.8%) adolescents had a combined biopsychosocial concept of pain, *n* = 45 (18.2%) had an exclusively psychosocial concept of pain, *n* = 38 (15.4%) had no pain concept, and *n* = 36 (14.6%) had a purely biological concept of pain. 

Similarly, most parents had a combined biopsychosocial concept of pain (*n* = 58; 48.3%), followed by *n* = 36 (30.0%) parents with a solely biological concept of pain, *n* = 19 (15.8%) parents with an exclusively psychosocial concept of pain, and *n* = 7 (5.8%) parents had no pain concept.

### 3.4. Multivariable Logistic Lasso Regression 

Lasso regressions were calculated for each outcome variable, using each factor rated as a potential cause for headaches as a predictor. The outcome variables that were significantly predicted by the factors were medication consumption and school absence (see Table 3). 

The only factor that significantly predicted high medication consumption was the factor “other disease” (OR = 3.15, 95% CI = 0.83–13.44, *p* = 0.041). The odds of adolescents who attributed another disease to their headache to take medication (more than) four days in the previous four weeks were 3.15 times as high as adolescents who did not rate another disease as causal for their headache. Moreover, the factor “organic cause” was retained in the model after variable selection but was nonsignificant. 

Moreover, school absence was significantly predicted by the factor “other disease” (OR = 3.40, 95% CI = 0.94–7.64, *p* = 0.03). Adolescents who considered the factor “other disease” causal for their headache were around three and a half times more likely of being (more than) two days absent from school compared to adolescents who did not rate “other disease” as a factor.

Lasso regressions that were calculated for the outcome variable pain-related disability did not reveal significant predictors. However, the factors school demands, puberty, other disease, and organic cause were all retained in the model. All variables retained in the model after variable selection are displayed in Table 3.

Factors that remained in the model after variable selection for the prediction of depressive symptoms were poor exercise, stress, high school demands, conflict with friends, nutrition, predisposition, other disease, organic cause, and conflict with family but were nonsignificant. Anxiety was not significantly predicted by any causal factor. 

## 4. Discussion 

The present study aimed to identify adolescents’ and parents’ causal explanations for adolescent headaches and to examine the concordance between them. Moreover, we examined the associations of these factors with behavioral and emotional outcomes. 

More than half of the adolescents and nearly half of the parents considered stress, caused by stressors not further specified, as one of the main factors causing the adolescents’ headaches. Several studies support this attribution as stress has been associated with primary headaches [21,40]. This association was also found in students, who often experienced primary headaches concurrently with elevated stress levels [41] and who were, consequently, more often absent from school [42]. In school children, school attendance seems to involve multiple important stressors (e.g., tests, personal expectations, grades) [43]. Accordingly, in our sample, school demands are the second most often rated psychosocial cause for headaches. To untangle the role of general stress and school-specific stress for the development of headaches, future research is needed.

Parents considered puberty as the second most causal headache attribute. The prevalence of pain increases during puberty for adolescent boys and girls [44]; however, the occurrence of primary headaches (e.g., migraine) significantly differs between girls and boys. This disparity may be linked to altered hormone levels [45,46] as well as structural and functional brain differences [47]. Boys tend to suffer from migraines at an earlier age, before puberty [48], while girls typically suffer the onset of headaches during puberty [44,49]. An increased frequency of migraine is primarily seen in postpubertal girls [50]. The high parental rating for puberty as an origin of headaches might also be due to certain behavioral changes in their children; an increase in conflicts between child and parent during puberty has been frequently observed [51]. A previous meta-analysis found that conflicts between parents and their children increase from early to mid-adolescence [52].

While agreement was fair regarding puberty, poor exercise, and conflicts with family, overall concordance between adolescents and parents was low. These results support previous studies that assessed concordance between parents and their children regarding pain experience in general [53,54]. However, our results contrast a study that reported rather high concordance between parents and adolescents in assessing the pain reports of adolescents with headaches [55]. This difference may be due to the study design. Our study focused on concordance regarding attribution of headache causes, while Kröner-Herwig et al. focused on agreement regarding headache characteristics and related symptoms. These results implicate the importance of consulting young patients about their explanatory models. However, asking the adolescents’ parents about their opinion might give additional insights into potentially causal factors.

When trying to investigate causes of pain, a biopsychosocial understanding of chronic pain is preferred to traditional biomedical approaches [56]. In our study, approximately half of adolescents with chronic headache and their parents were classified as having a biopsychosocial concept of pain. This shows that most adolescents and parents acknowledge that underlying biological causes lead to the experience of headaches, but that psychosocial factors also contribute. This outcome supports a study that assessed only mothers’ causal attributions of pain, which found that the majority endorsed a biopsychosocial model of pain [26]. In our study, the second most frequent pain concepts were purely psychosocial (adolescents) and exclusively biological (parents). A previous study found that teachers reported either a purely physical or psychosocial pain model [25]. This may be due to an unawareness of the child’s home environment, resulting in teachers attributing pain to only one domain, while the children and their parents have and use broad situational information for a more modern biopsychosocial understanding of pain [15,16]. However, the distinction made in this study between factors categorized as biological or psychosocial is debatable. The factor “puberty,” for example, was categorized as biological in this study due to the impact of hormonal changes; however, it might also be understood as psychosocial. For example, puberty may be associated with a more depressed mood and increased sensitivity, two aspects that are considered causal for headache occurrences. Moreover, certain biological and psychosocial factors may have a reciprocal association (e.g., exercise and media use).

Another major aim of this study was to examine the associations between factors of headaches and various outcomes. Our study revealed that medication consumption and days of school absence could be significantly predicted by a factor rated as causal for headaches. Interestingly, adolescents who chose the biological factor “other disease” as causing their headaches had greater odds of requiring more medication and of being more often absent from school than adolescents who did not choose this factor. Thus, designating a biological factor (anything the students understood as an “other disease”) was associated with behavioral outcomes. Although the lasso regression did not reveal significant predictors of the behavioral domain pain-related disability, several factors that were rated as causal remained in the model after variable selection. These were primarily the biological factors puberty, other disease, and organic cause. Thus, there might be an association between adolescents considering especially biological factors relevant to their headaches and behavioral outcomes. For the emotional domain, no factor significantly predicted any affective outcome variable. Therefore, it may be concluded that psychological impairments are not significantly associated with factors considered causal by the individual suffering from headache. However, since several factors remained in the model after variable selection for the prediction of depression there seems to be an association between these factors and depression, which cannot be further interpreted here but should be addressed in future investigations. More research is needed to support these findings and to investigate whether behavioral outcomes are indeed associated with biological factors considered causal, as well as whether the individual concept of pain is not significantly associated with emotional outcomes. 

Results of the present study need to be interpreted within the following limitations. First, 40.7% of data were missing for medication consumption; approximately 100 students could not discern the amount of medication they consumed in the previous four weeks. This may be a systematic issue, for example reflecting unattended self-medication, often found in adolescent samples [57] who do not pay attention to the amount they have taken. Another explanation might be that these adolescents do not check their medication consumption themselves but rely solely on their parents’ assessment of the amount of medication they take. However, the actual cause for these missing data in our study is unknown and this may have distorted the results of the logistic regression regarding the prediction of medication consumption. Another limitation is how the factors assessed in the checklist question about potential headache causes were described. The descriptions of these factors are vague (e.g., “stress” or “nutrition”). Thus, these factors need to be differentiated more specifically in future studies. Furthermore, the lasso regression confidence intervals and *p*-values should be interpreted with caution. It remains unclear how reliable these values are due to the bias introduced by penalizing the estimates [58]. Moreover, in the case of very small lambda values, estimations for postselection inferences are occasionally problematic. Therefore, several authors discourage reporting the lasso regression *p*-values and confidence intervals [58,59]. Instead, it is recommended to consider all variables retained in the lasso model after variable selection. 

This study showed that a subjective attribute of headaches influences behavioral outcomes. Therefore, future research should try to investigate the influence of factors seen as causal for chronic headache on the experience, treatment, and course of chronic headaches. Further insight into this topic is necessary to develop interventions for chronic headache, focusing on the influence of target factors.

## 5. Conclusions

This study is one of the first to assess adolescents’ own attribution of various factors of chronic headache. We found a lack of concordance between adolescents’ and parents’ opinions of what factors potentially caused headache pain. Thus, it seems essential to not rely exclusively on parental reports when assessing the origins of chronic pain in adolescents. Instead, the adolescent’s opinion about the origin of their headaches should be given more attention and listened to more carefully to avoid losing important information. Because the presumed causes of headache may be associated with behavioral outcomes, understanding individual concepts of pain should be addressed in future research. Further exploration of associations between the attributes of headache with the course and experience of chronic headaches might help to develop appropriate interventions for this vulnerable population.

## Figures and Tables

**Table 1 children-08-00234-t001:** Descriptive statistics of adolescents’ (*N* = 247) and parents’ (*N* = 120) judgments of potential causes of headaches.

	Adolescents (%)	Parents (%)	Kappa (κ) ^2^
Biological Factors ^1^			
Puberty	25.5	40.8	0.382
Other Disease	18.6	11.7	0.164
Predisposition	16.6	11.7	0.126
Injury	15.4	10.0	0.125
Nutrition	13.8	9.2	-0.032
Organic Cause	11.3	3.3	0.049
Poor Exercise	10.9	8.3	0.322
Psychosocial Factors ^1^			
Stress	53.4	42.5	0.092
School Demands	38.1	31.7	0.151
Media	22.7	19.2	0.179
Conflict with Friends	14.6	8.3	0.100
Conflict with Family	11.7	7.5	0.282
No Explanation ^1^	15.4	5.8	0.101

^1^ All factors were classified as “biological” or “psychosocial.” Subjects who chose no factor were categorized as “No explanation.” ^2^ Kappa values are interpreted as suggested by Landis and Koch (1977): 0 = poor agreement; 0–0.20 = slight agreement, 0.21–0.40 = fair agreement.

**Table 2 children-08-00234-t002:** Distribution of outcome variable categories for adolescents.

Outcome Variable	Frequency (N°)	Percentage (%)
Medication Use ≥4 days (past 4 weeks) (*N* = 147) ^1^		
No	111	75.0
Yes	36	25.0
School absence (past 4 weeks) (*N* = 237)		
<2 days	197	83.1
≥2 days	40	16.9
PPDI ^2^ (*N* = 247)		
Low pain-related disability	156	63.2
High pain-related disability	91	36.8
RCADS-Depression ^3^ (*N* = 248)		
No clinical symptoms of depression	152	61.3
Clinical symptoms of depression	96	38.7
RCADS-Anxiety ^3^ (*N* = 248)		
No clinical symptoms of anxiety	136	54.8
Clinical symptoms of anxiety	112	45.2

^1^ Data for medication use were only available for 147 adolescents. ^2^ Pediatric Pain Disability Index (PPDI). ^3^ Revised Children’s Anxiety and Depression Scale (RCADS).

**Table 3 children-08-00234-t003:** Factors associated with outcome variables after lasso variable selection.

	Behavioral Outcomes									Affective Outcomes		
	Medication Consumption			Pain-Related Disability ^1^			School Absence			Depression ^2^		
	OR	95%CI	*p*	OR	95%CI	*p*	OR	95%CI	*p*	OR	95%CI	*p*
Biological Factors												
Other disease	3.15	(0.83–13.44)	**0.041**	1.85	(0.55–3.61)	0.142	3.40	(0.94–7.64)	**0.030**	2.51	(0.30–5.03)	0.192
Organic cause	3.01	(0.01–8.57)	0.423	2.57	(0.72–5.83)	0.064				0.59	(0.23–1.51)	0.805
Poor Exercise										1.36	(0.00–11.34)	0.799
Predisposition										0.55	(0.15–3.70)	0.268
Nutrition										1.37	(0.00–53.97)	0.625
Puberty				1.46	(0.12–2.51)	0.514						
Psychosocial Factors												
Stress										0.58	(0.32–4.04)	0.317
School demands				1.71	(0.76–2.93)	0.084				1.80	(0.61–3.47)	0.127
Conflict with Friends										2.36	(0.73–6.33)	0.066
Conflict with Family										2.51	(0.71–7.55)	0.067

Significant predictors (*p* < 0.05) are displayed in bold. ^1^ Pain-related disability was measured by the Pediatric Pain Disability Index (PPDI). ^2^ Depression scores were extracted from the depression subscale of the Revised Children’s Anxiety and Depression Scale (RCADS).

## Data Availability

The data presented in the study are available on reasonable request from the corresponding author.

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
