# Peer review of "Adolescents’ Explanatory Models for Headaches and Associations with Behavioral and Emotional Outcomes"

_children, 2021, doi:10.3390/children8030234_

Round 1

Reviewer 1 Report

The overall design and intent are well founded. Overall it highlights that bio-psychological factors play a role in adolescent headache, and  that parent reporting is mildly discordant. Not a novel ideas.  However, this does ad depth of information to the field. 

A few questions about you they determined the binary measures of medication day use and school absences. Seems arbitrary and may be to improve there data breakdown. IF they could explain why they choose the breakdown.  

The definition of "other disease" is not clear to me. Is this true other disease. If so they should provide a breakdown of the diseases as this has bearing. If it the concern for possible other disease, then just clarify. 

Author Response

Dear Reviewer 1,

please see the attachment for our reply to your comment.

Kind regards,

Verena Neß

(In the name of all authors)

Reviewer 2 Report

This is a nice study  on a relevant topic, showing a perspective about parents and patients point of  view in headache triggers. This perspective is poorly described in the literature and in this study it was  investigated with a sound methodology.

See below some issues : 

Introduction . Line 42-43 The association between headache and Somatic Symptom Disorder should also be included.  Headache has been reported as one of the most common pain related to SSD in children admitted to a pediatric ED, see “  Somatic symptom disorder was common in children and adolescents attending an emergency department complaining of pain.Cozzi G, Acta Paediatr. 2017 .”  

Page 3 lines 114-115 The distinction between biological factors and psychosocial factor is somehow debatable : should poor physical exercise be considered a psychosocial factor due to isolation from peers , depression, poor motivation to engage in outdoor activities  ? This issue should be addressed in discussion.

Was obesity straightforwardly and explicitly  included  in the “nutrition” issue ?  The authors should better define how was this item coded. It was just a question “do you think nutrition is related to headache?”  

As a matter of fact, as it is in text, the issue of nutrition is not well  defined and difficult to understand and appreciate, even in the results section , in the multivariate analysis in table 3 referring to the association with depression . Does this mean that having a food intake related disorder was associated with depression ? Is it somehow related to overweight ?

Conclusions : line 337 . This statement is a bit too simple as it is. Could any physician imagine not to listen and speak to an adolescent complaining of headache ? I would rather put on emphasis on the need of an increased amount of listening in these patients , questioning the specific points .

Author Response

Dear Reviewer,

please see the attachment for our reply to your comment.

Kind regards,

Verena Neß

(In the name of all authors)

Reviewer 3 Report

Very interesting and relevant topic; there is often a significant disconnect between what parents think of their child's headache symptoms and triggers and what the patient thinks, so it is valuable to be able to quantify it to some extent.  

The missing medication consumption data is a limiting factor in the applicability of the data, as the missing 40.7% could significantly shift the other variables if it were known.  While this is addressed in the text, the explanation that "these adolescents trust in good parenting behavior..." (line 371-318) is unclear and should be reworded.  It would also be good to explore how the lack of data may have affected the other factors.

I found the psychosocial factors (listed on line 117: stress, high media consumption, conflicts in family environment, high school demands, and conflicts with friends and/or peers) to be somewhat confusing, as all the factors other than "stress" are clear causes of stressors.  Given that this is carried throughout the rest of the paper and "stress" is used as an independent variable, it would be good to clearly delineate what the terms covers, even if it is just to state that "stress" is a blanket term that covers undifferentiated stressors.  This is especially confusing in the discussion (line 254-255) where, in the section discussing stress, it is mentioned that adolescents rated several school-related items as frequent stressors when "high school demands" is already listed as a causal factor.

In line 14, "extend" is probably meant to be "extent." 

Author Response

(The authors gave the same response as above.)
